# Influence of competition on performance factors in under-19 soccer players at national league level

José María Izquierdo[1], Ana María De Benito[2], Gorka Araiz[1], Guillermo Guevara[1], Juan Carlos Redondo[1]*

1 Department of Physical Activity and Sport Sciences, University of León, León, Spain, 2 Faculty of Physical Activity and Sport Sciences, Catholic University of Valencia San Vicente Mártir, Valencia, Spain

* jc.castan@unileon.es

**Data Availability Statement:** All relevant data are within the manuscript and its Supporting Information files.

## Abstract

The aim of this study was to analyse and quantify the acute effects of competition on several performance factors in under-19 male soccer players. To this end, 198 national league players (17.56 ± 0.78 years) performed various tests to measure jump capacity, kicking velocity and sprint times immediately pre-match (T1), at half-time (T2) and post-match (T3). Tests included kicking the ball to measure ball velocity (KICK), sprinting for 40 meters, timing the first 30 meters (30mACCEL), the last 10 meters (10mACCEL) and the total distance (40mACCEL), and performing countermovement jumps (CMJ). For subsequent analysis, the sample was divided into 5 playing positions: goalkeepers (n = 24), defenders (n = 51), midfielders (n = 36), wingers (n = 54) and forwards (n = 33). For all positions, we found a significant decline in performance (p<0.05) for kicking velocity (2.91% - 6.51%) and sprinting (0.44%-5.85%). For the CMJ, all positions except defenders presented a significant decline in performance that ranged from 1.5% to 4.56%. These findings highlight the need to individualise fitness training, taking into account the match needs and demands of the different playing positions in order to minimise the effects of match fatigue and accelerate post-match recovery.

## Introduction

As a discontinuous or acyclic team sport, soccer requires dynamic, random and intermittent actions [1] involving acceleration, jumping and changes of direction and speed, all of which exerts great pressure on neuromuscular and metabolic parameters [2]. Sprint times and vertical jump capacity have been used to assess soccer players' anaerobic profile [3] and to study changes with respect to the beginning of the match [4].

It has been demonstrated that soccer matches cause physical changes in players that lead to reduced physical performance in aspects such the capacity to maintain high speed during sprints [5], and acceleration/deceleration distances during and after the match [6]. Moreover, male players' jump and knee extension-flexion performance declines immediately after a

**Funding:** The authors received no specific funding for this work.

**Competing interests:** The authors have declared that no competing interests exist.

soccer match [7], and player performance is associated with various factors such as body structure, strength, power and speed [8]. However, the changes these factors undergo do not affect all playing positions equally [1].

Soccer players execute multiple sprints during a match [9], and although these represent less than 10% of the total distance covered, they are nonetheless decisive in the match outcome [10]. Frequent acceleration and deceleration during a soccer match involves high energy consumption and induces momentary fatigue in players [11]. As a result, maximum sprint speed declines over the course of the match [4], hence the capacity to minimise fatigue is related to an increase in performance [12]. The need to execute sprints varies during a match, thus soccer players must be capable of sprinting, recovering and sprinting again at the highest intensity possible [9].

Furthermore, other high intensity acyclic actions are necessary during a match, such as kicking, jumping, braking and explosive starts, and these vary according to playing position. Malina et al. [13] have reported significant differences in vertical jump, speed over 30 meters and intermittent aerobic endurance between defenders, midfielders and forwards in a group of young elite players with 4.5 years of experience. It has also been demonstrated [14] that forwards obtain better results for the 30 meter sprint and vertical jump than other playing positions. In that sense, the acute effect of fatigue on jumps post-match has also been demonstrated in other sports such as handball [15] and rugby [16], but not in basketball [17].

Many researchers consider kicking a fundamental aspect of a soccer player's performance because it is the most frequently employed action during a match [18], for example when attempting to score a goal or passing the ball to other players. When kicking, players aim to achieve different ball velocities and trajectories with a high degree of accuracy [19]. Thus, some studies have analysed this variable to assess kicking accuracy, using a radar to measure kicking velocity [20].

Acceleration, the vertical jump and kicking velocity could be considered among the most easily measured acyclic actions due to the simplicity of their execution and ease of reproduction. Although these actions exert a decisive influence on performance in soccer, their variations have not been extensively studied in a competitive context. Consequently, the aim of the present study was to analyse and quantify the acute effects of a soccer match on the performance of under-19 players, testing jump capacity, sprint times and kicking velocity pre-match, at half-time and post-match, contrasting results according to playing position. In view of the above, we hypothesized that match fatigue would cause a clear reduction in the values obtained for the variables analysed in all players expecting changes in results regarding the position of the players.

## Materials and methods

We used a repeated measures design to assess the effects of regular soccer matches on various performance factors. In order to determine the influence of accumulated match fatigue, the study was conducted during the last third of the playing season, analysing official federation matches. This study was conducted in accordance with the guidelines found in the Helsinki Declaration which establishes ethical principles for investigations using human beings. and was approved by the Ethics Committee of the University of Leon.

### Participants

Participants comprised one hundred ninety-eight healthy male soccer players with a mean age of 17.56 ± 0.78 years old and a mean of 9.50 ± 3.11 years' experience of playing in federation soccer teams. Participants belonged to four different Spanish youth clubs of the same

Table 1. Descriptive statistics (mean ± standard deviation).

|  | All players (n = 198) | Goalkeepers (n = 24) | Defenders (n = 51) | Midfielders (n = 36) | Wingers (n = 54) | Forwards (n = 33) |
|---|---|---|---|---|---|---|
| Age (years) | 17.56±0.78 | 17.21±0.84 | 17.58±0.77 | 17.55±0.87 | 17.71±0.72 | 17.57±0.72 |
| Height (cm) | 178.09±5.63 | 178.00±4.81 | 180.76±5.88 | 178.50±6.96 | 174.72±3.88 | 179,19±3.80 |
| Weight (kg) | 69.03±7.84 | 71.13±5.43 | 71.71±6.72 | 70.75±10.13 | 65.22±6.73 | 67.76±7.42 |
| Body Mass Index (kg/m²) | 21.74±2.10 | 22.47±1.81 | 21.96±1.96 | 22.18±2.91 | 21.34±1.86 | 21.05±1.57 |
| Experience (years) | 9.51±3.12 | 8.38±3.37 | 10.18±2.02 | 9.67±2.20 | 8.83±3.83 | 10.24±3.54 |

geographic area (Castilla y León, Spain), with same chronological ages and competing in the same youth Spanish national category with regular training. All subjects were in good health and were not taking medication or nutritional supplements that could influence the experimental protocol. Participants were divided into five groups based on playing position and player characteristics: goalkeepers (n = 24), defenders (n = 51), midfielders (n = 36), wingers (n = 54) and forwards (n = 33). Demographic and anthropometric data are shown in Table 1. Study participation was voluntary. All players, parents and coaches were notified of the research procedures, requirements, benefits and risks before giving written informed consent.

## Procedure

Testing procedures were performed during competition seasons 2016–2017 and 2017–2018. Prior to data collection at matches, players were familiarised with the tests by performing them during training. The tests were then conducted at thirty-three different matches, played in the second part of each analysed season. We selected players who would play for the maximum number of minutes, identified in a prior conversation with the coach. To ensure efficiency and interfere as little as possible in match preparation, the tests were conducted with six players for each match, including at least one player for each of the five positions analysed. To standardise procedure, all tests were performed using the same protocol and in the same order before, during and after the match. As can be seen in Fig 1, the tests were performed after pre-match

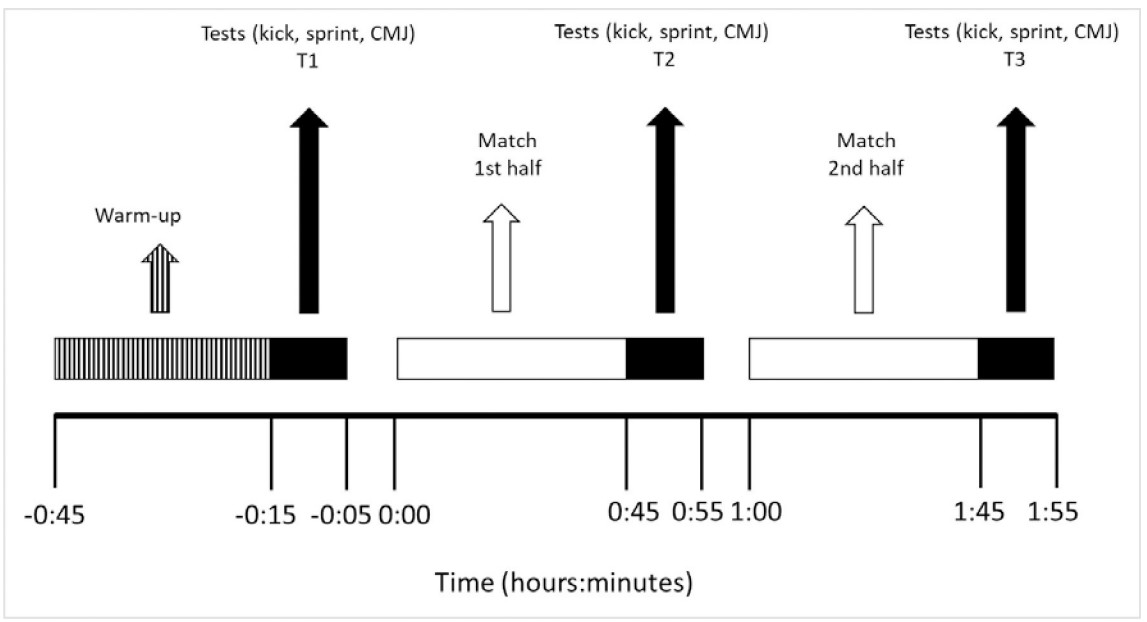

Fig 1. Timing and order of tests.

warm-up (T1), at half-time (T2) and post-match (T3). If a player was substituted, he performed the tests at that time provided that he had completed at least 50% of the second half. All matches were played on natural grass surface and no were register data of matches in rainy days. The average temperature was between 8˚C and 20 ºC (data obtained from the Spanish Meteorological Agency www.aemet.es)

**Kicking velocity.**   Kicking velocity was measured as the maximum ball velocity when aiming at the goal, according to the protocol described by Sedano et al. [21]. Velocity was expressed in kilometers per hour (km/h), measured using a Stalker´s type hyperfrequency radar (Stalker Professional Radar, Radar Salts, Plymouth, MA, USA). The players made 3 attempts with 60 seconds' rest between kicks, and the best result in km/h was used to analyse the KICK variable.

**Acceleration capacity.**   Acceleration capacity was assessed by means of a 40 m sprint test conducted in an adjoining field with an identical surface to that where the match was played, and timed using a single beam photocell system (DSD Laser System, León, Spain). Two photocells were sited at the start, another two at 30 m and two at 40 m. Players started the sprint test half a meter behind the first two photocells [22]. A visual stimulus was used to indicate to players when the cells were ready to start, and then players could start, on their own, a 40-m sprint at maximum intensity. They performed this test once, and times for the first 30 m (30mACCEL), the last 10 m (10mACCEL) and the total 40 m (40ACCEL) were analysed.

**Explosive force-jump capacity.**   Players' jump capacity was measured using a laser platform (SportJUMP System PRO, DSD Inc., Spain), placed in a small area adjacent to the team dressing room with a surface similar to that of the match surface. Keeping their hands at the waist, players made 3 attempts at the CMJ [23], with 60 seconds' rest between jumps. The best result in centimeters was analysed for the CMJ variable.

**Statistical analysis.**   Data analysis was carried out using the Statistical Package for Social Sciences (SPSS 24.0). The Kolmogorov-Smirnov test (K-S) was used to determine normal distribution of the variables analysed. Standard statistical methods were used to calculate means and standard deviations (SD). The Student's t-test was used to determine differences between initial values for the 5 groups and the variables analysed. Effects related to the match were assessed using one-way ANOVA (time) with repeated measures, post Levene's test for equality of variances (p>0.05). When Wilks' Lambda indicated a significant F-value, Bonferroni post-hoc tests were performed to determine pairwise differences. Statistical significance was set at p<0.05. The effect size (ES) was estimated using Cohen's *d* [24], classifying ES as "small" (0.2–0.3), "medium" (0.4–0.7) and "large" (>0.8).

## Results

The Kolmogorv-Smirnov test confirmed that the variables presented a normal distribution (p>0.05). The Student's *t*-test indicated that there were no significant differences between groups at baseline (p>0.05). Table 2 shows the data for the variables analysed at the three times (T1, T2 and T3) and Table 3 gives the same variables for each playing position.

### Kicking velocity

For the variable KICK, we found significant differences between times ($p = 0.00$; ES = 0.396). The Bonferroni post-hoc test identified differences between T1 and T2 ($p = 0.00$; ES = 0.285) and T3 ($p = 0.00$; ES = 0.758) and between T2 and T3 ($p = 0.00$; ES = 0.510). As regards differences by playing position (Fig 2), the Bonferroni post-hoc test identified the following significant differences: in goalkeepers between T1 and T2 ($p = 0.033$; ES = 0.167) and T3 ($p = 0.006$; ES = 0.488) and between T2 and T3 ($p = 0.033$; ES = 0.351); in defenders between T1 and T2

**Table 2. Descriptive data for the variables kicking velocity (KICK), acceleration (ACCEL) and jump capacity (CMJ).**

| | T1 | T2 | T3 | Time |
|---|---|---|---|---|
| **Variables** | **Mean ± SD** | **Mean ± SD** | **Mean ± SD** | **$F$ (ES)** |
| **KICK (km/h)** | 97.83±6.16 | 96.17±5.44 | 93.31±5.74 | 71.003* (0.448) |
| **30mACCEL (s)** | 4.68±0.24 | 4.75±0.33 | 4.82±0.25 | 51.092* (0.349) |
| **40mACCEL (s)** | 5.95±0.31 | 6.02±0.28 | 6.16±0.31 | 63.156* (0.396) |
| **10mACCEL (s)** | 1.26±0.10 | 1.28±0.16 | 1.33±0.11 | 41.323* (0,306) |
| **CMJ (cm)** | 35.72±4.68 | 34.65±4.79 | 34.14±4.78 | 41.281* (0.301) |

* $p < .05$

($p = 0.00$; ES = 0.390) and T3 ($p = 0.00$; ES = 0.829) and between T2 and T3 ($p = 0.00$; ES = 0.468); in midfielders between T1 and T2 ($p = 0.00$; ES = 0.444) and T3 ($p = 0.00$; ES = 1.038) and between T2 and T3 ($p = 0.00$; ES = 0.678); in wingers between T1 and T3 ($p = 0.00$; ES = 0.571) and between T2 and T3 ($p = 0.001$; ES = 0.401); and in forwards between T1 and T3 ($p = 0.00$; ES = 0.983) and between T2 and T3 ($p = 0.00$; ES = 0.771).

**Table 3. Descriptive data and percentages of variation for the variables kicking velocity (KICK), acceleration (ACCEL) and jump capacity (CMJ), by playing position.**

| | | T1 | T2 | T3 | Percentage of variation | | |
|---|---|---|---|---|---|---|---|
| **Variables** | **Position** | **Mean ± SD** | **Mean ± SD** | **Mean ± SD** | **T1-T2** | **T1-T3** | **T2-T3** |
| **KICK (km/h)** | GOALKEEPER | 93.43$_a$±7.28 | 92.29$_b$±6.14 | 90.14$_c$±5.88 | -1.22% | -3.52% | -2.32% |
| | DEFENDER | 97.87$_a$±6.01 | 95.60$_b$±5,52 | 92.93$_c$±5.80 | -2.32% | -5.04% | -2.79% |
| | MIDFIELDER | 99.18$_a$±6.42 | 96,55$_b$±5,27 | 92.72$_c$±5.86 | -2,66% | -6.51% | -3.95% |
| | WINGER | 97.00$_a$±4.66 | 96.12$_a$±4.45 | 94.18$_b$±5.14 | -0.91% | -2.91% | -2.02% |
| | FORWARD | 100.71$_a$±5.86 | 99.29$_a$±4.90 | 95.22$_b$±5.62 | -1.41% | -5.44% | -4.09% |
| **30mACCEL (s)** | GOALKEEPER | 4.85±0.21 | 4.87±0.19 | 4.91±0.21 | -0.49% | -1.32% | -0.82% |
| | DEFENDER | 4.68$_a$±0.22 | 4.88$_b$±0.50 | 4.88$_b$±0.23 | -4.16% | -4.14% | 0.03% |
| | MIDFIELDER | 4.76$_a$±0.24 | 4.74$_a$±0.25 | 4.88$_b$±0.23 | 0.44% | -2.40% | -2.85% |
| | WINGER | 4.59$_a$±0.22 | 4.65$_b$±0.25 | 4.79$_c$±0.31 | -1.47% | -4.42% | -2.90% |
| | FORWARD | 4.61$_a$±0.24 | 4.63$_a$±0.15 | 4.71$_b$±0.16 | -0.64% | -2.11% | -1.46% |
| **40mACCEL (s)** | GOALKEEPER | 6.15±0.27 | 6.17$_a$±0.29 | 6.22$_b$±0.31 | -0.24% | -1.12% | -0.87% |
| | DEFENDER | 6.00$_a$±0.31 | 6.08$_a$±0.27 | 6.23$_b$±0.29 | -1.28% | -3.77% | -2.46% |
| | MIDFIELDER | 6.04$_a$±0.34 | 6.06$_a$±0.31 | 6.22$_b$±0.31 | -0.33% | -3.08% | -2.74% |
| | WINGER | 5.82$_a$±0.27 | 5.95$_b$±0.28 | 6.10$_c$±0.38 | -2.33% | -4.80% | -2.42% |
| | FORWARD | 5.83$_a$±0.24 | 5.88$_a$±0.16 | 6.03$_b$±0.15 | -0.92% | -3.37% | -2.42% |
| **10mACCEL (s)** | GOALKEEPER | 1.3±0.12 | 1.29±0.12 | 1.31±0.14 | 0.64% | -0.38% | -1.03% |
| | DEFENDER | 1.29$_a$±0.09 | 1.31$_a$±0.11 | 1.34$_b$±0.12 | -1.26% | -4.12% | -2.83% |
| | MIDFIELDER | 1.27$_a$±0.12 | 1.32$_b$±0.08 | 1.35$_b$±0.09 | -3.20% | -5.60% | -2.32% |
| | WINGER | 1.23$_a$±0.07 | 1.3$_b$±0.08 | 1.33$_b$±0.12 | -5.85% | -7.75% | -1.79% |
| | FORWARD | 1.25$_a$±0.07 | 1.26$_a$±0.05 | 1.32$_b$±0.07 | -1.21% | -6.13% | -4.86% |
| **CMJ (cm)** | GOALKEEPER | 35.26$_a$±7.26 | 34.39$_a$±7.30 | 33.42$_b$±8.01 | -2.48% | -5.22% | -2.80% |
| | DEFENDER | 35.32±3.56 | 34.79±4.41 | 34.88±3.17 | -1.50% | -1.23% | 0.27% |
| | MIDFIELDER | 33.77±5.74 | 32.44±5.03 | 31.81±4.13 | -3.92% | -5.79% | -1.94% |
| | WINGER | 35.87$_a$ ±3.62 | 34.23$_b$ ±3.68 | 33.35$_b$ ±4.56 | -4.56% | -7.02% | -2.58% |
| | FORWARD | 38.59±2.46 | 37.69±2.60 | 37.36±2.59 | -2.33% | -3.17% | -0.86% |

Means on the same line with the same subscript do not present significant differences at $p < 0.05$.

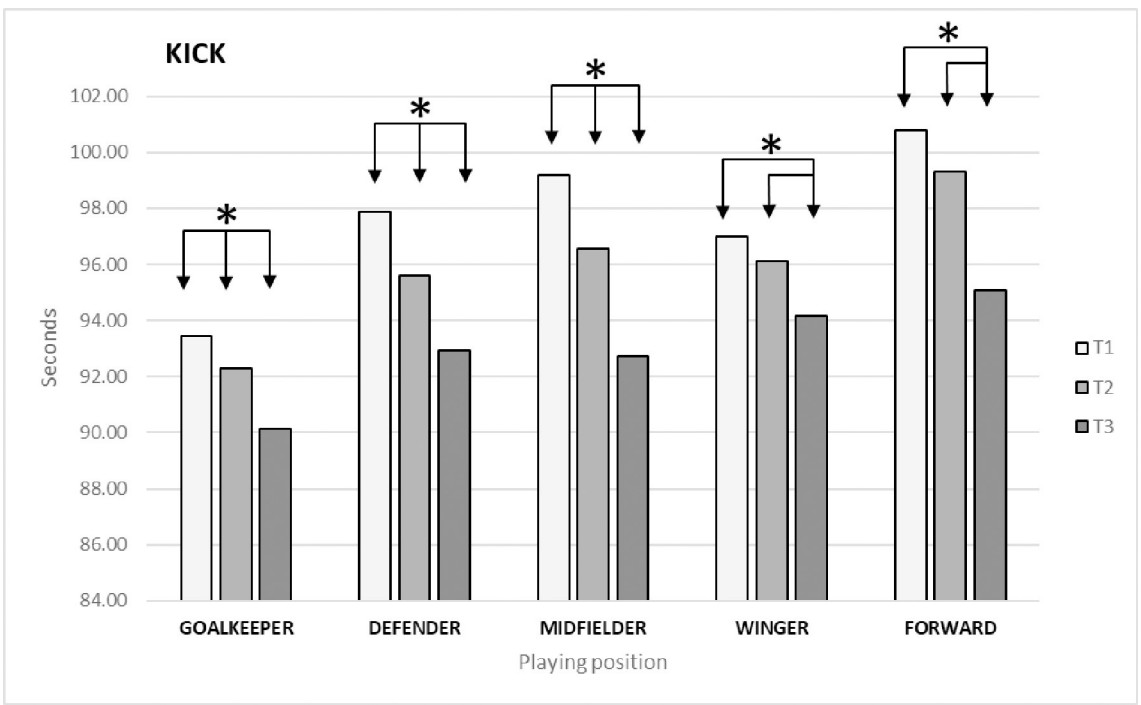

**Fig 2. Changes in kicking velocity (KICK) by playing position.**

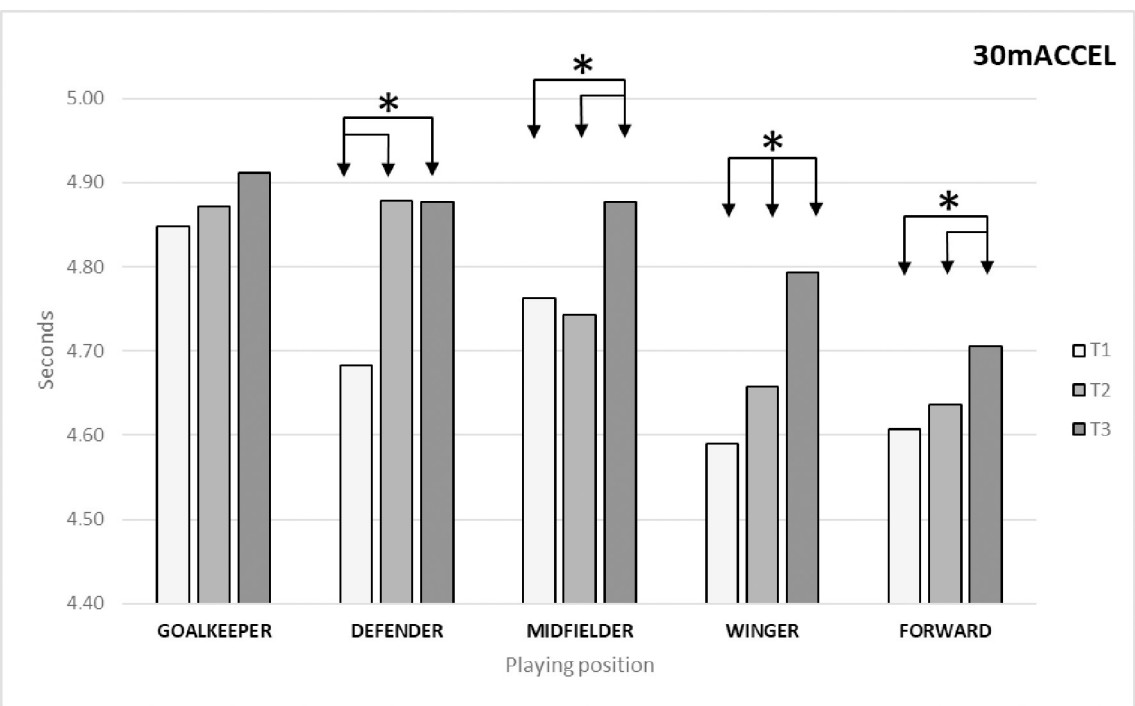

**Fig 3. Changes in acceleration capacity (30mACCEL) by playing position.**

## Acceleration capacity

As regards acceleration capacity, Table 2 shows that the variable 30mACCEL presented significant differences between times ($p$ = 0.00; ES = 0.301). The Bonferroni post-hoc test identified differences between T1 and T2 ($p$ = 0.00; ES = 0.305) and T3 ($p$ = 0.00; ES = 0.570). As regards differences in this variable by playing position (Fig 3), the Bonferroni post-hoc test identified the following significant differences: in defenders between T1 and T2 ($p$ = 0.026; ES = 0.502) and T3 ($p$ = 0.00; ES = 0.859); in midfielders between T1 and T3 ($p$ = 0.00; ES = 0.483) and between T2 and T3 ($p$ = 0.00; ES = 0.484); in wingers between T1 and T2 ($p$ = 0.003; ES = 0.248) and T3 ($p$ = 0.00; ES = 0.753) and between T2 and T3 ($p$ = 0.00; ES = 0.474); and in forwards between T1 and T3 ($p$ = 0.017; ES = 0.481) and between T2 and T3 ($p$ = 0.015; ES = 0.384).

For the variable 40mACCEL(Table 2), we found significant differences between times ($p$ = 0.00; ES = 0.396). The Bonferroni post-hoc test identified differences between T1 and T2 ($p$ = 0.00; ES = 0.236) and T3 ($p$ = 0.00; ES = 0.676) and between T2 and T3 ($p$ = 0.00; ES = 0.451). As regards differences by playing position (Fig 4), the Bonferroni post-hoc test identified the following significant differences: in goalkeepers between T2 and T3 ($p$ = 0.014; ES = 0.177); in defenders between T1 and T2 ($p$ = 0.017; ES = 0.263) and T3 ($p$ = 0.00; ES = 0.748) and between T2 and T3 ($p$ = 0.00; ES = 0.535); in midfielders between T1 and T3 ($p$ = 0.00; ES = 0.569) and between T2 and T3 ($p$ = 0.00; ES = 0.474); in wingers between T1 and T2 ($p$ = 0.00; ES = 0.493) and T3 ($p$ = 0.00; ES = 0.838) and between T2 and T3 ($p$ = 0.00; ES = 0.220); and in forwards between T1 and T3 ($p$ = 0.00; ES = 0.968) and between T2 and T3 ($p$ = 0.00; ES = 0.914).

As shown in Table 2, the variable 10mACCEL presented significant differences between times ($p$ = 0.00; ES = 0.306). The Bonferroni post-hoc test identified differences between T1 and T3 ($p$ = 0.00; ES = 0.664) and between T2 and T3 ($p$ = 0.00; ES = 0.363). As regards

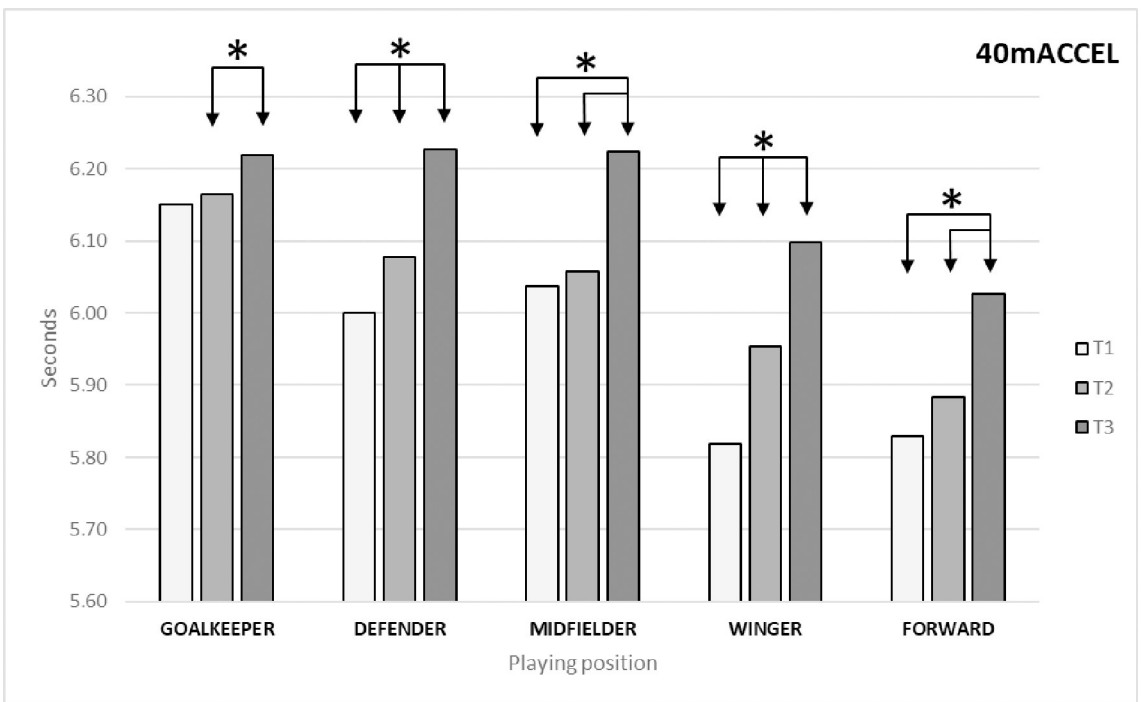

**Fig 4. Changes in acceleration capacity (40mACCEL) by playing position.**

differences by playing position ([Fig 5]), the Bonferroni post-hoc test identified the following significant differences: in defenders between T3 and T1 ($p = 0.001$; ES = 0.489) and T2 ($p = 0.002$; ES = 0.317); in midfielders between T1 and T2 ($p = 0.016$; ES = 0.395) and T3 ($p = 0.00$; ES = 0.664); in wingers between T1 and T2 ($p = 0.00$; ES = 0.908) and T3 ($p = 0.00$; ES = 0.915); and in forwards between T1 and T3 ($p = 0.00$; ES = 1.061) and between T2 and T3 ($p = 0.00$; ES = 0.152).

### Explosive force- jump capacity

As shown in [Table 2], the results revealed significant differences between times ($p = 0.00$; ES = 0.301) for the variable CMJ. The Bonferroni post-hoc test identified differences between T1 and T2 ($p = 0.00$; ES = 0.093) and T3 ($p = 0.00$; ES = 0.657).

As regards differences between times by playing position ([Fig 6]), the Bonferroni post-hoc test identified the following significant differences: in goalkeepers between T1 and T2 ($p = 0.007$; ES = 0.104) and T3 ($p = 0.00$; ES = 0.209) and between T2 and T3 ($p = 0.009$; ES = 0.109); in midfielders between T1 and T3 ($p = 0.001$; ES = 0.339); in wingers between T1 and T2 ($p = 0.00$; ES = 0.388) and T3 (p = 0.00; ES = 0.531) and between T2 and T3 ($p = 0.00$; ES = 0.185); and in forwards between T1 and T2 ($p = 0.029$; ES = 0.309) and T3 ($p = 0.003$; ES = 0.421).

### Discussion

Due to the paucity of studies conducted in competitive contexts, the aim of the present study was to analyse the acute effects of a soccer match on jump capacity, sprint times and kicking velocity in under-19 players. We expected to find a general decline in performance due primarily to muscle fatigue occasioned by the demands of the match [25]. We also expected to

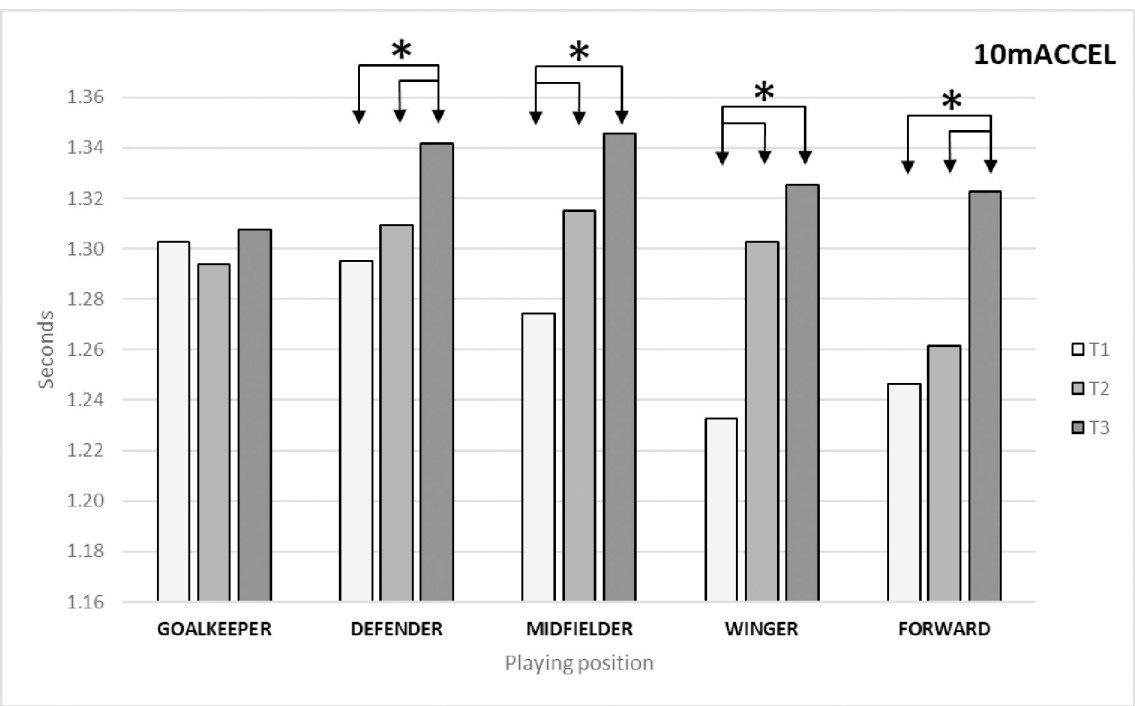

**Fig 5. Changes in acceleration capacity (10mACCEL) by playing position.**

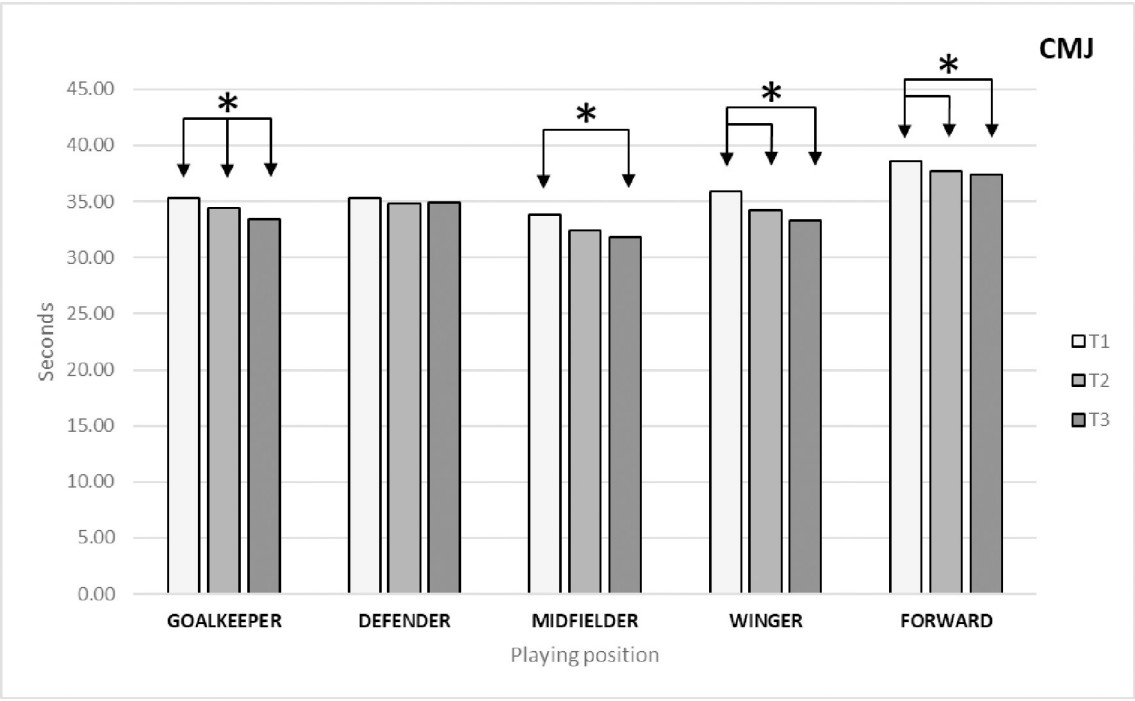

**Fig 6. Changes in explosive force (CMJ) by playing position.**

find that the effect of this fatigue would vary between players according to playing position, since each position involves specific features and functions [1]. Both hypotheses were confirmed by our study. We found that performance declined over time for all the variables analysed, and also that this was associated with playing position, in under-19 soccer players.

In a literature review, Rodríguez Lorenzo et al. [26] reported values for the kicking velocity test according to age, sex, dominant leg, experience, competition level and playing position, and found differences between defenders and midfielders and forwards. In contrast to these findings, our results indicated that kicking velocity was similar for all playing positions except goalkeepers, with forwards obtaining slightly higher values.

With respect to the reduction in kicking velocity caused by match fatigue, it has been reported that accuracy, kick success rate and maximum ball velocity decline significantly in under-19 players in the second half [27]. This finding is consistent with our results, since we observed a statistically significant decline in performance between the start and end of the match and between half-time and the end of the match for all playing positions. It therefore seems clear that fatigue exerts a negative effect on kicking velocity by reducing players' capacity to transfer velocity from leg to ball [28], possibly due to an increase in lactate production and a reduction in glycogen levels, which also affects coordination [10]. However, some studies have reported that kicking velocity is affected by body posture, kicking technique, boots or a reduced ability to transfer energy to the ball due to loss of muscle strength [29]. Studies of young players [30] have also found that good hamstring flexibility exerts an important influence not only on kicking velocity, but also on jump capacity and sprint times.

Sander et al. [31] related vertical strength training programs to improved sprint times in young soccer players. Along the same lines, a marked decline in CMJ and sprint performance has been reported, indicating that several aspects of the capacity to generate strength are compromised in male and female soccer matches alike [32]. In this study, it was found a significant

decline in CMJ and sprint performance, although the above studies assessed the 20-m sprint rather than the 30-m sprint analysed here, and did not differentiate by playing position. Thus, it could be inferred that in the case of the CMJ, changes related to muscle fatigue become evident immediately after the match and are long-lasting [33], while in the case of the sprint, the different rates of recovery between the sprint and the jump may be related to changes in biomechanical behaviour (the duration of concentric and eccentric phases), resulting in neuromuscular fatigue that affects jump performance to a greater extent [33].

Carlin et al. [34] have argued that speed tests may be more appropriate than jump tests to assess performance and fatigue. Recently, it has been demonstrated that performance of a short sprint (10–20 m) in a straight line lacks sensitivity as an indicator of physical fatigue immediately post-match [35]. Such a short distance does not alter times but does affect hip biomechanics, with changes in flexion and extension angles [36]. Nagahara et al. [37] have suggested using running distances in excess of 30 m in order to improve assessment of the effect of post-match fatigue on soccer players' maximum speed capacity, and obtained differences in this capacity (p = 0.038) when they used a distance of 35 m. However, this effect of fatigue has also been demonstrated using distances of up to 60 m [35]. Our results for all playing positions except goalkeepers and all distances (10, 30 and 40 m) indicate a statistically significant decline in performance between the start and end of the match and between half-time and the end. The reason for the lack of such a difference could be considered to result from the nonexistence of the repeated sprints because of the position of goalkeepers during the match [10]. However, even though the distance of 10 m has been studied in other sports, such as basketball [17], it is important to emphasise that these results cannot be compared with our findings since our distance of 10 m was measured at the end of the 40-m test (between 30 m and 40 m) and therefore represented a full speed.

Assessing jump capacity in soccer players by means of the CMJ test it has also proved a reliable indicator of fatigue in soccer players [33]. The fatigue produced over the course of a match was evident in all players analysed in our study except for the defenders. In this sense, sprint times and vertical jump capacity are related as aspects to assess soccer players' anaerobic profile [3], and defenders spend a significantly less amount of time sprinting and running than the other positions [1]. In a study by Romagnoli et al. [38], players of a similar age to those in our study obtained a better result (49.6±5.1 cm pre-match) than our participants (35.72 cm), but the decline in their performance was similar to the mean obtained in our study. In addition, there is an acute effect on the isometric strength of the associated hamstrings during knee flexion [39], a basic movement in jump technique. This post-match fatigue remains significant 24 hours post- match for this test [7], and in most cases there is no functional recovery until the 72 hours post-match [35, 38]. However, Stone et al. [35] found no significant differences in this decline in their study. It should be noted that age could also be a factor in this test, because although the effect of fatigue has been elucidated in young players [40]. In other sports such as handball [15] and rugby [16], post-match fatigue has been shown to affect jump capacity, no significant decline has been observed.

Although it is acknowledged that there are a number of limiting factors which may account for the present findings, we believe that the present design has high levels of validity and demonstrates the need for further research to continue identifying and quantifying the physical requirements—and their inter-relationships—of different playing positions. In that sense, must be taken into consideration that our sample not investigate how nature of the game (importance of the game or superiority of the opponent) or weather conditions could influence the measurement results. Furthermore, it would be beneficial to explore other player profiles, levels and categories, such as professional or female soccer players, and to study these effects at different times of the playing season, because in the final months of the competitive mesocycle

(a decisive phase in the season), the cumulative load of several months of training and competition may exert a different effect to that in other periods.

## Conclusions and practical applications

In this study, we have quantified the decline in performance of tests of jump capacity, sprint times and kicking velocity produced as a result of the demands of a soccer match in under-19 players. This decline in the performance of the tests did not affect all playing positions equally, demonstrating the need for individualised fitness training.

Our results provide useful information for coaches and trainers with respect to the organisation of training. In order to attain good physical performance, it is necessary to minimise the effects of match fatigue and accelerate post-match recovery. This can be achieved by tailoring training strategies and programs to each of the players' needs.

## Supporting information

**S1 Data. Supporting information.**
(XLSX)

## Acknowledgments

The authors wish to thank the athletes who participated in the study for their effort and dedication, and their coaches and clubs for their help and availability.

## Author Contributions

**Conceptualization:** José María Izquierdo, Juan Carlos Redondo.

**Investigation:** José María Izquierdo, Ana María De Benito, Gorka Araiz, Guillermo Guevara.

**Methodology:** Juan Carlos Redondo.

**Supervision:** José María Izquierdo.

**Validation:** Ana María De Benito.

**Writing – original draft:** Ana María De Benito, Juan Carlos Redondo.

**Writing – review & editing:** Juan Carlos Redondo.

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
