## [Decision Letter · Decision Letter 0]

2 Jan 2020

PONE-D-19-31381

Influence of competition on performance factors in under-19 soccer players at national league level

PLOS ONE

Dear Dr. Redondo Castán,

Thank you for submitting your manuscript to PLOS ONE. After careful consideration, we feel that it has merit but does not fully meet PLOS ONE’s publication criteria as it currently stands. Therefore, we invite you to submit a revised version of the manuscript that addresses the points raised during the review process.

We would appreciate receiving your revised manuscript by Feb 16 2020 11:59PM. To enhance the reproducibility of your results, we recommend that if applicable you deposit your laboratory protocols in protocols.io, where a protocol can be assigned its own identifier (DOI) such that it can be cited independently in the future. For instructions see: http://journals.plos.org/plosone/s/submission-guidelines#loc-laboratory-protocols

We look forward to receiving your revised manuscript.

Kind regards,

Dragan Mirkov, Ph.D.

Academic Editor

PLOS ONE

Additional Editor Comments:

Please read carefully the Reviewers comments and try to address it particularly regarding the statistical analysis.

2. In your Methods section, please provide additional information about the participant recruitment method and the demographic details of your participants. Please ensure you have provided sufficient details to replicate the analyses such as: a) the recruitment date range (month and year), b) a description of any inclusion/exclusion criteria that were applied to participant recruitment, c) a table of relevant demographic details, d) a statement as to whether your sample can be considered representative of a larger population, e) a description of how participants were recruited, and f) descriptions of where participants were recruited and where the research took place.

3. Please provide additional details regarding participant consent.

In the ethics statement in the Methods and online submission information, please ensure that you have specified (1) whether consent was informed and (2) what type you obtained (for instance, written or verbal).

As your study included minors, state whether you obtained consent from parents or guardians.

If the need for consent was waived by the ethics committee, please include this information.

'The funders had no role in study design, data collection and analysis, decision to publish, or preparation of the manuscript.'

Please provide an amended Funding Statement that declares *all* the funding or sources of support received during this specific study (whether external or internal to your organization) as detailed online in our guide for authors at http://journals.plos.org/plosone/s/submit-nowPlease state what role the funders took in the study.  If any authors received a salary from any of your funders, please state which authors and which funder. If the funders had no role, please state: "The funders had no role in study design, data collection and analysis, decision to publish, or preparation of the manuscript."

Reviewers' comments:

Reviewer's Responses to Questions

**Comments to the Author**

1. Is the manuscript technically sound, and do the data support the conclusions?

Reviewer #1: Partly

Reviewer #2: Yes

2. Has the statistical analysis been performed appropriately and rigorously? 

Reviewer #1: No

Reviewer #2: Yes

3. Have the authors made all data underlying the findings in their manuscript fully available?

Reviewer #1: Yes

Reviewer #2: Yes

4. Is the manuscript presented in an intelligible fashion and written in standard English?

Reviewer #1: Yes

Reviewer #2: Yes

5. Review Comments to the Author

Reviewer #1: This study aims to examine the influence of competition on performance factors in under-19 soccer players at national league level. The study is quite intriguing, however there are some serious methodological flaws, that can not be neglected. The manuscript is rather well written, although some improvement on that field is also needed. Please find general and specific comments below:

General comments:

1. Introduction needs some thorough revision, regarding writing style. From the perspective of potential readers, sentences seem to be hard to read and connect. It appears to me they are just tossed one after another. Some fluency is needed. Authors should “tell us a story” about this topic (based on the previous research, of course).

2. Since the players were observed by position in team, descriptive data (age, height, weight, experience) should be presented by their position as well (with appropriate statistics to show potential differences). Moreover, BMI data would also be beneficial for this manuscript.

3. One of the major methodological flaws in this manuscript is the fact that the authors tested players in different occasions during the extensive period of time. Great number of factors can influence this methodological design, such as time of a day, different weather conditions (temperature, humidity, wind...), nature of the game (important game of not, better or inferior opponent...), type of the surface (some football fields have better quality of grass, thus allowing greater speeds and better results)... Overall, great number of factors was not controlled by the researchers. Therefore, results could not be accepted such as.

4. This manuscript is lack of study limitations paragraph within the discussion chapter.

Specific comments:

Page 4, Lines 74-75: Hypothesis should be more informative, regarding players’ position. Do you expect changes in results regarding the position of the players or not?

Page 5, Lines 87-89: Can participants give consent by themselves, since majority of them are underage?

Page 6, Line 117: Did they use visual stimulus as a signal to start immediately, or that the cells are ready, and they can start on their own?

Page 6, Line 123: Please specify, how can surface in dressing room be similar to the one on the football field?

Page 5, procedure: I did not noticed explanation for test order in procedure. Authors should specify that.

Page 7, statistical analysis: Leven test should be performed prior to the all ANOVAs, especially since there are unequal numbers of participants within the groups.

Page 7, Lines 134-135: Can authors be more precise? Is Cohen’s coefficient, Cohen's D or something else?

Page 7, Line 140: Please present p level.

Page 11, Line 219: In introduction, only one hypothesis is presented. Please, make sure that aims and hypothesis are the same in abstract, introduction, discussion and conclusion.

Page 11, Line 224: Please change “forward.”

Page 11-12, Lines 227-238: Authors should specifically discuss why there are differences between goalkeepers and other players.

Page 12, Line 242: Please use “in this study” and “it was found”.

Page 12-13, Discussion: Authors did not provide sufficient explanation regarding differences between positions. Discussion was primarily focused on overall decline in performance.

Page 14, Line 280: Please avoid terms such as “interesting”. For example, say: “it would be beneficial (for coaches, scientists) to explore”…

Reviewer #2: This is very interesting work. Easy for reading and good in all methodical procedures. It is a specially interesting for sport sciences and of course for football coaches. Of course it will be interesting that in work we have more date about number of sprinting, jumps and acceleration and deceleration and changing of direction of every player in games.

6. PLOS authors have the option to publish the peer review history of their article (what does this mean?). If published, this will include your full peer review and any attached files.

Reviewer #1: No

Reviewer #2: No

---

## [Author Response · Author response to Decision Letter 0]

16 Jan 2020

By the academic editor

- It was revised.

2. In your Methods section, please provide additional information about the participant recruitment method and the demographic details of your participants. Please ensure you have provided sufficient details to replicate the analyses such as: 

a) the recruitment date range (month and year). 

- In page 5, this paragraph was modified (lines 100-103):

Testing procedures were performed during competition seasons 2016-2017 and 2017-2018. Prior to data collection at matches, players were familiarised with the tests by performing them during training. The tests were then conducted at thirty-three different matches, played in the second part of each analysed season.

b) a description of any inclusion/exclusion criteria that were applied to participant recruitment

- In page 5, this paragraph was added (lines 88-91):

Participants belonged to four different Spanish youth clubs of the same geographic area (Castilla y León, Spain), with same chronological ages and competing in the same youth Spanish national category with regular training. All subjects were in good health and were not taking medication or nutritional supplements that could influence the experimental protocol.

c) a table of relevant demographic details

- Table 1 was updated including data by positions and IMC 

d) a statement as to whether your sample can be considered representative of a larger population, 

- Considering that the category involves 14 teams in competition and that a total of 182 matches are played, we can regard our sample (33 matches) as representative a priori. Which is confirmed by the normality of the sample and the significance of the differences found in the parametric tests. All this, reinforced by the size of the calculated effect.

e) a description of how participants were recruited

It is the same that answered in section b: Participants belonged to four different Spanish youth clubs of the same geographic area (Castilla y León, Spain)

f) descriptions of where participants were recruited and where the research took place. 

- In page 5, this paragraph was added (lines 88-91):

Participants belonged to four different Spanish youth clubs of the same geographic area (Castilla y León, Spain), with same chronological ages and competing in the same youth Spanish national category with regular training.

3. Please provide additional details regarding participant consent.

In page 5, lines 95-96 were added:

- All players, parents and coaches were notified of the research procedures, requirements, benefits and risks before giving written informed consent.

In the ethics statement in the Methods and online submission information, please ensure that you have specified (1) whether consent was informed and (2) what type you obtained (for instance, written or verbal).

- In page 4, this paragraph (lines 82-84) was added:

This study was conducted in accordance with the guidelines found in the Helsinki Declaration which establishes ethical principles for investigations using human beings. and was approved by the Ethics Committee of the University of Leon.

It was an error. There are no restrictions and we upload our data set necessary to replicate our study findings.

- Included in the cover letter:

The authors received no specific funding for this work.

 

By reviewer #1:

1. Introduction needs some thorough revision, regarding writing style. From the perspective of potential readers, sentences seem to be hard to read and connect. It appears to me they are just tossed one after another. Some fluency is needed. Authors should “tell us a story” about this topic (based on the previous research, of course).

We have thoroughly reviewed the introduction with the reader in mind proceeding from the general to the particular. We have tried to link ideas, so it is easier for the reader to understand the structure adding connectors such as furthermore, hence, however or thus.

2. Since the players were observed by position in team, descriptive data (age, height, weight, experience) should be presented by their position as well (with appropriate statistics to show potential differences). Moreover, BMI data would also be beneficial for this manuscript.

In page 5, table 1 was updated including data by positions and BMI

3. One of the major methodological flaws in this manuscript is the fact that the authors tested players in different occasions during the extensive period of time. Great number of factors can influence this methodological design, such as time of a day, different weather conditions (temperature, humidity, wind...), nature of the game (important game of not, better or inferior opponent...), type of the surface (some football fields have better quality of grass, thus allowing greater speeds and better results)... Overall, great number of factors was not controlled by the researchers. Therefore, results could not be accepted such as.

We believe this consideration is very important. First, we made it clear that no match data was recorded on rainy days, although it was not a condition taken into account a priori, since the photocells and radar are not prepared for rain. On the other hand, the AEMET (Spanish Meteorological Agency) database was consulted and the temperature range of the days and time of the matches was established.

So, this paragraph was included in lines 110-113 (page 6):

All matches were played on natural grass surface and no were register data of matches in rainy days. The average temperature was between 8oC and 20oC (data obtained from the Spanish Meteorological Agency www.aemet.es)

Regarding the “nature of the game”, since it has not been included in the study variables, a comment is attached to its limitations.

Added in lines 294-304 (page 14), changing the existing paragraph:

Although it is acknowledged that there are a number of limiting factors which may account for the present findings, we believe that the present design has high levels of validity and demonstrates the need for further research to continue identifying and quantifying the physical requirements —and their inter-relationships— of different playing positions. In that sense, must be taken into consideration that our sample not investigate how nature of the game (importance of the game or superiority of the opponent) or weather conditions could influence the measurement results. Furthermore, it would be beneficial to explore other player profiles, levels and categories, such as professional or female soccer players, and to study these effects at different times of the playing season, because in the final months of the competitive mesocycle (a decisive phase in the season), the cumulative load of several months of training and competition may exert a different effect to that in other periods.

4. This manuscript is lack of study limitations paragraph within the discussion chapter.

Specific comments:

Page 4, Lines 74-75: Hypothesis should be more informative, regarding players’ position. Do you expect changes in results regarding the position of the players or not?

In page 4, lines 76-77 was added:

…expecting changes in results regarding the position of the players.

Page 5, Lines 87-89: Can participants give consent by themselves, since majority of them are underage?

In page 5, lines 95-96 were added:

All players, parents and coaches were notified of the research procedures, requirements, benefits and risks before giving written informed consent.

Page 6, Line 117: Did they use visual stimulus as a signal to start immediately, or that the cells are ready, and they can start on their own?

In pages 6 and 7, lines 128-129 were modified:

A visual stimulus was used to indicate to players when the cells were ready to start, and then players could start, on their own, a 40-m sprint at maximum intensity.

Page 6, Line 123: Please specify, how can surface in dressing room be similar to the one on the football field?

Page 7, line 134, this paragraph was modified: 

“placed in a small area adjacent to the team dressing room”

Page 5, procedure: I did not noticed explanation for test order in procedure. Authors should specify that.

This point it exposed in Fig 1, cited in page 6: 

Page 7, statistical analysis: Leven test should be performed prior to the all ANOVAs, especially since there are unequal numbers of participants within the groups.

Data analysis was carried out using the Statistical Package for Social Sciences (SPSS 24.0) and Leven test is included the protocol of ANOVA calculations.

So we have included in line 139. Data analysis was carried out using the Statistical Package for Social Sciences (SPSS 24.0).

Added in line 144: post Levene's test for equality of variances (p >0.05).

Page 7, Lines 134-135: Can authors be more precise? Is Cohen’s coefficient, Cohen's D or something else?

It is analized the Cohen’s d effect sizes related to Cohen J. Statistical power analysis for the behaviors science.(2nd). New Jersey: Laurence Erlbaum Associates, Publishers, Hillsdale. 1988.

So we have rewritten the sentence in lines 146-148:

The effect size (ES) was estimated using Cohen’s d [24], classifying ES as “small” (0.2-0.3), “medium” (0.4-0.7) and “large” (>0.8). 

Page 7, Line 140: Please present p level.

It was added (p>0.05) in line 144

Page 11, Line 219: In introduction, only one hypothesis is presented. Please, make sure that aims and hypothesis are the same in abstract, introduction, discussion and conclusion.

In page 4, lines 76-77 was added so this aspect has been corrected:

…expecting changes in results regarding the position of the players 

Page 11, Line 224: Please change “forward.”

We don’t understand what to do. The word forwards in included in a cite of Rodríguez Lorenzo et al.

Page 11-12, Lines 227-238: Authors should specifically discuss why there are differences between goalkeepers and other players.

When we have review our results (page 9, lines 161-170) we realized that there are no differences between goalkeepers and other players so we have modified this aspect (line 242) and we believe it is no necessary to discuss about this. 

Page 12, Line 242: Please use “in this study” and “it was found”.

It was changed (Line 253).

Page 12-13, Discussion: Authors did not provide sufficient explanation regarding differences between positions. Discussion was primarily focused on overall decline in performance.

At this regard it has been added the next paragraphs:

Page 13, Lines 271-273:

The reason for the lack of such a difference could be considered to result from the nonexistence of the repeated sprints because of the position of goalkeepers during the match [10].

Pages 13-14, Lines 280-283:

In this sense, sprint times and vertical jump capacity are related as aspects to assess soccer players’ anaerobic profile [3], and defenders spend a significantly less amount of time sprinting and running than the other positions [1].

Page 14, Line 280: Please avoid terms such as “interesting”. For example, say: “it would be beneficial (for coaches, scientists) to explore”…

It was changed (Line 300)

---

## [Decision Letter · Decision Letter 1]

21 Feb 2020

Influence of competition on performance factors in under-19 soccer players at national league level

PONE-D-19-31381R1

Dear Dr. Redondo Castán,

We are pleased to inform you that your manuscript has been judged scientifically suitable for publication and will be formally accepted for publication once it complies with all outstanding technical requirements.

With kind regards,

Dragan Mirkov, Ph.D.

Academic Editor

PLOS ONE

Additional Editor Comments (optional):

Reviewers' comments:

Reviewer's Responses to Questions

**Comments to the Author**

1. If the authors have adequately addressed your comments raised in a previous round of review and you feel that this manuscript is now acceptable for publication, you may indicate that here to bypass the “Comments to the Author” section, enter your conflict of interest statement in the “Confidential to Editor” section, and submit your "Accept" recommendation.

Reviewer #1: All comments have been addressed

Reviewer #2: All comments have been addressed

2. Is the manuscript technically sound, and do the data support the conclusions?

Reviewer #1: Yes

Reviewer #2: Yes

3. Has the statistical analysis been performed appropriately and rigorously? 

Reviewer #1: Yes

Reviewer #2: Yes

4. Have the authors made all data underlying the findings in their manuscript fully available?

Reviewer #1: Yes

Reviewer #2: Yes

5. Is the manuscript presented in an intelligible fashion and written in standard English?

Reviewer #1: Yes

Reviewer #2: Yes

6. Review Comments to the Author

Reviewer #1: All raised issues, comments and questions have been thoroughly addressed by the authors of this paper.

Reviewer #2: This study aims to examine the influence of competition on performance factors in under-19 soccer players at national league level. I still think that this is very interesting work. Easy for reading and good in all methodical procedures. It is a specially interesting for sport sciences and of course for football coaches.

7. PLOS authors have the option to publish the peer review history of their article (what does this mean?). If published, this will include your full peer review and any attached files.

Reviewer #1: No

Reviewer #2: No

---

## [Editor Report · Acceptance letter]

6 Mar 2020

PONE-D-19-31381R1 

Influence of competition on performance factors in under-19 soccer players at national league level 

Dear Dr. Redondo:

I am pleased to inform you that your manuscript has been deemed suitable for publication in PLOS ONE. Congratulations! Your manuscript is now with our production department. 

With kind regards,

on behalf of

Dr. Dragan Mirkov 

Academic Editor

PLOS ONE